# Criteria for and Policy Implications of Setting Recovery Priorities of National Functions during Disruptions by Disasters

Seok Bum Hong [1], Jin Byeong Lee [2], Jeong Hoon Shin [1] and Hong Sik Yun [1,2,*]

1   Interdisciplinary Program for Crisis, Disaster and Risk Management, Sungkyunkwan University, Suwon 16419, Republic of Korea; how098@naver.com (S.B.H.); hoon6433@google.com (J.H.S.)
2   School of Civil and Architectural Engineering, Sungkyunkwan University, Suwon 16419, Republic of Korea; ljb3897@naver.com
*   Correspondence: yoonhs@skku.edu

**Abstract:** National functions are categories of operations prioritised for restoration when disrupted by emergencies such as disasters. However, the simultaneous restoration of all national functions when some or all are paralysed is limited by time and resources. Delays in the restoration of key functions can lead to public dissatisfaction. Thus, it is necessary to broadly classify national functions and analyse their restoration priorities based on criticality. This study identifies 19 national functions from Republic of Korea's comprehensive Business Reference Model. A survey was conducted among citizens and officials to determine the criticality of each function. Statistical analyses verified the consistency (Cronbach's alpha = 0.860) and correlation (average Cramer's $V$ = 0.107) of the criticality responses across regions. The null hypothesis of no regional differences in the criticality of national functions was accepted, validating their universality. Restoration priorities were derived from these criticality values, with 'Disaster Safety Response' as the highest priority and 'Regional Development' as the lowest. These results provide foundational data for the post-disaster restoration priorities of national functions and emphasise the need to consider public opinion, needs, and government resource limitations in disaster management planning.

**Keywords:** major national functions; recovery priority; BRM; MAO

## 1. Introduction

Disasters are evolving and are projected to become more frequent and intense. A nation comprises multiple 'national functions', which can be disrupted by intensifying disasters. These functions can be categorised under various national institutions, including central administrative agencies, public organisations, research institutions, and power plants. Disaster-induced interruption of these functions, which deliver high-quality public services, results in significant economic and social damage. The government is responsible for restoring disruptions to national functions within a permissible timeframe after disasters. However, both the human and financial resources at government disposal are limited [1].

Societal resilience comprises four main components: robustness, redundancy, resourcefulness, and rapidity. Of these, resourcefulness pertains to diagnosing problems, establishing priorities, and effectively utilising resources, whereas rapidity denotes the capability of swift restoration through prioritisation to minimise the impact of external shocks [2]. There is a pressing need to prioritise and swiftly restore functions with high criticality to enhance resourcefulness and rapidity, the pivotal components of resilience. Establishing a recovery priority for national functions necessitates surveying and analysing public-demand levels and evaluating the criticality of each function. The failure to rapidly restore key national functions can exacerbate societal dissatisfaction. Pertinent examples include the following.

The Gyeongju earthquake on 12 September 2016 was among the most powerful earthquakes experienced in the country. The government, spearheaded by the Ministry of Public Administration and Safety, collaborated with other relevant agencies to undertake recovery efforts. However, the inadequate restoration of evacuation facilities (function classification: Disaster Safety Response) and emergency medical facilities (function classification: Health Services) drew significant criticism. Additionally, many victims expressed dissatisfaction with the insufficiency of state support.

On 11 March 2011, northeastern Japan was struck by an earthquake of magnitude 9.0, resulting in power-supply challenges at the Fukushima Daiichi nuclear plant. This situation escalated into a crisis marked by the evaporation of cooling water from the plant and a subsequent hydrogen explosion within the reactor core. The catastrophe disrupted essential functions in multiple sectors, including industry, health, and transportation. The cessation of nuclear power production, categorised under the 'Industry' function, resulted in power shortages across several regions in Japan. Owing to the impact on the 'Health Services' and 'Transportation' functions, there was a minimum of 789 casualties on the day of the disaster, including in-transit fatalities caused by transportation disruptions.

Hurricane Katrina struck New Orleans on 29 August 2005. Despite the need for swift infrastructure restoration and financial support for evacuees, the government's response was inadequate. Consequently, timely access to necessary services was denied to many, and the resulting social and economic damage continued to mount.

Several researchers from Republic of Korea and other countries have performed classification and criticality analyses of national functions. In the South Korean context, studies have derived disaster responses and resilience-related functions and analysed their criticality and prioritisation [3,4]. Regarding international cases, studies have analysed and prioritised impediments delaying recovery [5]. These studies ranked disaster-response functions based on survey analyses. However, they did not conduct a comprehensive analysis of all national functions. Several studies have investigated the detailed functional classification system of the South Korean Business Reference Model and proposed improvements [6–8] in national functions, including culture and communication. However, their analysis of the recovery prioritisation of these functions is inadequate.

Studies have explored the significance and improvement of specific subfunctions to enhance disaster resilience. For instance, one study investigated resilience based on a model of collaboration among national organisations [9,10]. Similarly, a few studies proposed directions for enhancing disaster resilience through improving disaster-related infrastructure [11]. Another study analysed the impact of architectural design techniques on disaster resilience [12]. Based on international case studies, researchers have examined the significance of disaster-risk governance in countries such as Croatia, Greece, and Nepal [13–16]. In Australia, one study investigated the impact of the health protection capacity of local governments on disaster resilience [17]. These studies have emphasised the criticality of specific subfunctions. However, they have limitations in terms of analysing the recovery priorities of these functions.

Several studies have attempted to derive and analyse functions to understand disaster resilience [18,19]. One study identified functions such as governance, resources, risk reduction, funding acquisition, and time compression for the effective analysis of long-term housing recovery frameworks [20]. Another study derived 24 Disaster Preparedness Indicators and conducted a survey analysis of disaster preparedness awareness among university students [21]. Similarly, a survey-based study elucidated the functions affecting disaster resilience in the wake of storms and analysed their relative criticalities [22]. Furthermore, numerous studies have identified and analysed functions for disaster-risk reduction and enhancing disaster resilience [23–25]. While these studies have delineated and analysed functions for assessing disaster resilience, their analyses of national functions beyond disaster management are limited.

An examination of research on establishing contingency plans in the event of a disruption to specific group functions found studies that examined strategic approaches to

minimise risk exposure during supply chain interruptions [26]. These studies discussed resource prioritisation, but did not focus specifically on disasters; they also had the limitation of not providing concrete recommendations. Additionally, a few studies have proposed methodologies for universally applicable business continuity plans [27,28], facilitating our understanding of the business continuity plan, but they lack content on the prioritisation of function recovery.

The Business Impact Analysis within the business continuity plan is a procedure that identifies critical functions and processes of an organisation. In addition, it evaluates the potential impacts should these functions or processes be interrupted. From this perspective, several studies have proposed methodologies to identify key functions in the public sector and assess their potential impact in the event of their disruption [29,30]. This study referenced these methodologies to analyse and prioritise the significance of national functions and their impact on citizens.

Related manuals and research cases on the classification and criticality analysis of national functions were identified. One Japanese disaster-response guideline, 'Guide to Business Continuity for Local Public Entities During Earthquake Occurrence and Its Commentary', describes the method for determining the start target time for emergency functions—one of the 'national functions' mentioned in the Introduction [31]. Chapter 3 of the Japanese government's 'Central Agency Business Continuity Guidelines' discusses the content pertaining to priority tasks and administrative affairs during disasters. Although these guidelines define regular tasks separately from emergency priority tasks, they have the limitation of not quantitatively analysing the priority of these tasks [32].

A literature review revealed that a few studies and manuals had partially analysed 'national functions' or categorised them with a focus on disasters. However, no study has classified all national functions or analysed recovery priorities based on their criticality. Previous research has only partially addressed national functions in disaster scenarios. This study extends this by comprehensively analysing all national functions using the South Korean Business Reference Model with a focus on their classification and recovery prioritisation. A nationwide survey assessed the criticality of national functions, yielding a recovery priority model for efficient resource utilisation during disasters. This marks a significant innovation in national function classifications and recovery strategies. Therefore, this study aims to conduct a survey and analysis of the requirement levels of national functions, target the aforementioned South Korean Business Reference Model, and develop a model for recovery priority. As aforementioned, various local self-governing bodies and public institutions operating in Republic of Korea can also be categorised under multiple 'functions'. The government classifies these functions in a Business Reference Model that includes a systematic, comprehensive categorisation of national functions. A national survey was conducted to collect data to analyse the criticality of 19 national functions selected from the Business Reference Model. The consistency of the survey results was ascertained using Cronbach's alpha. In addition, the frequency of responses was examined to determine the criticality of each function. Cramer's *V* analysis was used to verify the significant differences in criticality across regions. Finally, the recovery priorities for national functions were derived based on the verified requirement levels. The resulting recovery priority is expected to serve as foundational data for a strategy to facilitate the more efficient use of limited resources when national functions are interrupted by a disaster.

## 2. Materials and Methods

The international regulation regarding the establishment of the recovery priorities for 'national functions' is described under the functional analysis stage of the Continuity of Operations Plan (COOP) within International Organization for Standardization (ISO) 22301 [33,34]. The functional analysis stage states that all national functions should be categorised. There is a need to analyse the priorities of functions that must be urgently resumed in the event of an emergency [35]. The overall research flow is shown in Figure 1. The consistency of the respondents' answers was analysed using Cronbach's alpha, and,

in accordance with the necessities defined in the COOP, 19 national functions were extracted from Republic of Korea's Business Reference Model and their criticality analysed through a nationwide survey. The results were compared using Cramer's *V* analysis to ascertain the national consistency in criticality across regions. The recovery priorities of the national functions with verified consistency were derived based on their average criticality. This can serve as a criterion when establishing recovery priorities for major functions in disaster situations.

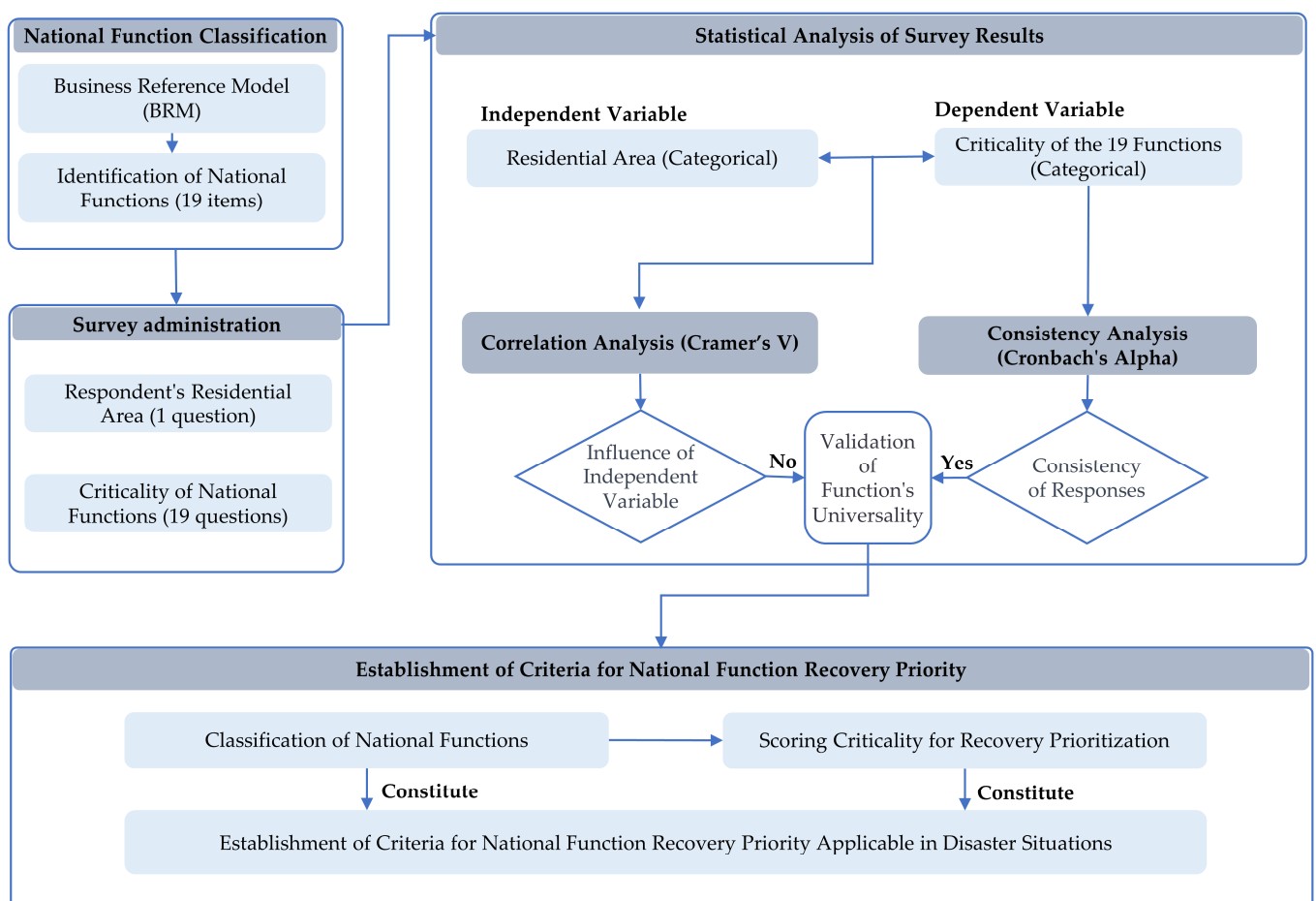

**Figure 1.** Research flow for deriving the criteria of major national functions' recovery priority.

## 3. Survey Implementation and Statistical Analysis

### 3.1. Classification of Functions Using the Business Reference Model

National functions represent the functions of local autonomous bodies when conducting local administration. A local autonomous body is 'a self-governing entity formed by residents living in a specific region to handle politics and administration of the local community independently from the central government' [36]. National functions (alternatively, 'local administrative functions'), defined from the perspective of local autonomous bodies, are 'functions aimed at realising the public interest of the local community while being regulated by law'. Republic of Korea's Local Autonomy Law specifies the legal classification of the administrative functions of South Korean local autonomous bodies". Based on this legal foundation, Republic of Korea utilises a Business Reference Model for national and local classification.

Based on the Public Records Management Act as revised in 2007, the South Korean government introduced the Business Reference Model as a record-management system to systematically manage government tasks. Initiated by central administrative agencies in 2008, it has been adopted by various public institutions, including local autonomous bodies

and educational offices, and serves as a vital classification system. There are two types of government function-classification systems: 'functional classification', the categorisation of ongoing governmental functions based on legal systems; and 'objective-based classification', a system primarily organised around the policy objectives of each department. As this study focuses on the government's functions, the 'objective-based classification' that addresses the ongoing 'functions' of the government has been applied. This classification of functions comprises five stages: Policy Sector (Level 1), Policy Domain (Level 2), Primary Function (Level 3), Intermediate Function (Level 4), and Minor Function (Level 5). This study primarily conducted surveys using the topmost level, Policy Sector [37]. Table 1 presents the composition of the government's function-classification system. To generalise the functions in Table 1, it is necessary to explore whether the functions addressed by Republic of Korea's Business Reference Model are applicable globally.

**Table 1.** List and details of national functions based on the business reference model.

| Survey No. | National Functions | | Details | Class |
|---|---|---|---|---|
| Q1 | | Recovery | Operation of disaster recovery funds, support for recovery activities, etc. | Primary Function |
| Q2 | | Response | Disaster firefighting, rescue, medical treatment activities, etc. | Primary Function |
| Q3 | Disaster Safety | Preparation | Regular safety policy, disaster safety education, disaster situation management, etc. | Primary Function |
| Q4 | | Prevention | Development of disaster safety technology, preventive management, etc. | Primary Function |
| Q5 | Public Order and Safety | | Police, legal affairs, prosecution, and societal public order and safety maintenance | Policy Sector |
| Q6 | Science and Technology | | Creation of a scientific and technological research environment and technological development | Policy Sector |
| Q7 | Education | | Compulsory education for infants/elementary/middle school, higher education, lifelong/vocational education, etc. | Policy Sector |
| Q8 | National Defence | | Military conscription, mobilisation, military administration, etc. | Policy Sector |
| Q9 | Agriculture, Forestry, Marine, and Fisheries | | Agriculture, livestock, food farmland repair, rural development, and distribution of agricultural and marine products | Policy Sector |
| Q10 | Culture, Sports, and Tourism | | Promotion of culture, sports, and tourism, and preservation of cultural heritage | Policy Sector |
| Q11 | Health Services | | Health, sanitation, disease prevention, food and drug safety management, and operation of emergency medical systems | Policy Sector |
| Q12 | Social Welfare | | Social security, social services, and promotion of social welfare | Policy Sector |
| Q13 | Industry and SMEs | | Industrial technology support, trade and investment attraction for industrial promotion, energy and resource development | Policy Sector |
| Q14 | Transportation and Logistics | | Management of transportation logistics systems (roads, railways, aviation, ports, maritime, logistics) | Policy Sector |
| Q15 | Public Administration | | General administration, finance, taxation, and financial management | Policy Sector |
| Q16 | Regional Development | | Development of industrial zones and regional development | Policy Sector |
| Q17 | Telecommunications | | Information communication, broadcasting, and post-office management | Policy Sector |
| Q18 | Unification and Diplomacy | | Diplomacy, trade negotiations, treaties, other international agreements, international situation investigation, immigration, and unification management | Policy Sector |
| Q19 | Environmental Protection | | Air, water quality, waste, natural environment, and environmental pollution management | Policy Sector |

Based on the research presented in this study, the approach to setting post-disaster response priorities is primarily grounded in the national context of Republic of Korea and Maslow's hierarchy of needs, and yet it exhibits a high degree of universality. The 19 index

elements derived from comparing policies of several major countries broadly align with fundamental human needs, suggesting their applicability beyond the specific national conditions of Republic of Korea.

To evaluate the universality of this study's findings, we compared the national functions classification systems established in other major countries with the domestic Business Reference Model (Table 2). The United States follows the National Critical Functions Set [38] in classifying national functions, which can be completely matched with all 19 functions extracted from Republic of Korea's business reference models. Japan has divided its classifications corresponding to the United States' operational domains into three major categories: public administrative services, operational execution methods, and internal administrative tasks. Furthermore, they delineated 23, 8, and 6 task classifications within these categories, respectively, with 88, 30, and 34 respective sub-classifications [39].

**Table 2.** List of primary national functions of major countries similar to Republic of Korea's national functions.

| Survey No. | Republic of Korea | | USA | Japan | EU |
|---|---|---|---|---|---|
| Q1 | | Recovery | | | |
| Q2 | Disaster | Response | Prepare for and Manage | Disaster Management | Disaster Management |
| Q3 | Safety | Preparation | Emergencies | | |
| Q4 | | Prevention | | | |
| Q5 | Public Order and Safety | | Provide Public Safety | Public Safety and Social Order | Law and Justice |
| Q6 | Science and Technology | | Manufacture Equipment | Development and Science | Energy and Technology |
| Q7 | Education | | Educate and Train | Education and Training | Education and Workforce |
| Q8 | NationalDefence | | Provide Materiel and Operational Support to Defence | Defence | Defence and Security |
| Q9 | Agriculture, Forestry, Marine, and Fisheries | | Produce and Provide Agricultural Products and Services | Agriculture, Forestry, and Fisheries | Agricultural Affairs |
| Q10 | Culture, Sports, and Tourism | | Provide and Maintain Infrastructure | Promotion and Public Relations | - |
| Q11 | Health Services | | Provide Medical Care | Public Health | Health and Well-Being |
| Q12 | Social Welfare | | Provide Insurance Services | Social Activity Support | Employment, Social Affairs, and Inclusion |
| Q13 | Industry and SMEs | | Produce Chemicals | Industrial Support | Internal Market, Industry, Entrepreneurship, and SMEs |
| Q14 | Transportation and Logistics | | Transport Cargo and Passengers | Transportation | Transport and Space |
| Q15 | Public Administration | | Develop and Maintain Public Works and Services | Public Asset Management | General Government |
| Q16 | Regional Development | | Provide Housing | - | Economic and Financial |
| Q17 | Telecommunications | | Maintain Supply Chains | Energy/Resource Management | Communications |
| Q18 | Unification and Diplomacy | | Enforce Law | Diplomacy | Diplomacy and Trade |
| Q19 | Environmental Protection | | Manage Hazardous Materials | Environmental Management | Environment and Natural Resources |

Additionally, the European Union (EU) has established the European Interoperability Reference Architecture, which comprises 10 main categories, each further subdivided. This architecture can be juxtaposed and compared with Republic of Korea's Business Reference Model [40]. This study conducted a comparative analysis between Republic of Korea's Business Reference Model and the national function-classification systems of the United States, Japan, and the EU. The U.S. 'National Critical Functions Set' matched all functions of

the South Korean model, while Japan and the EU's models showed similar structures. This international comparison underscores the high universality and excellence of Republic of Korea's Business Reference Model in the national function classification. Table 2 compares Republic of Korea to other advanced countries with similar national systems. Function matching revealed the universality of the 19 functions extracted from the South Korean Business Reference Model.

### 3.2. Survey Structure and Maximum Acceptable Outage

Investigating the public-demand level for the 19 functions selected from the Business Reference Model represents the initial step essential for analysing recovery priorities. A survey was conducted among 1506 adults aged 20 years and older residing nationwide (public servants: 123; non-public servants: 1383). Figure 2 illustrates the regional distribution of the respondents. Seoul had the highest number of respondents (446), while Jeju-do had the lowest (14). The survey was conducted in 2018, before the outbreak of the coronavirus disease 2019 (COVID-19) pandemic. Such pandemics differ from traditional emergencies because of their global impact, potentially causing shifts in the significance of national functions. However, research based on pre-COVID-19 data enabled us to discern the significance of various national functions during typical non-pandemic emergencies. Although the global repercussions of the pandemic persist, there has been a gradual return to normalcy [41], warranting the use of pre-pandemic data and ensuring ongoing applicability. By leveraging survey data from 2018, this study aimed to analyse the recovery priorities of national functions.

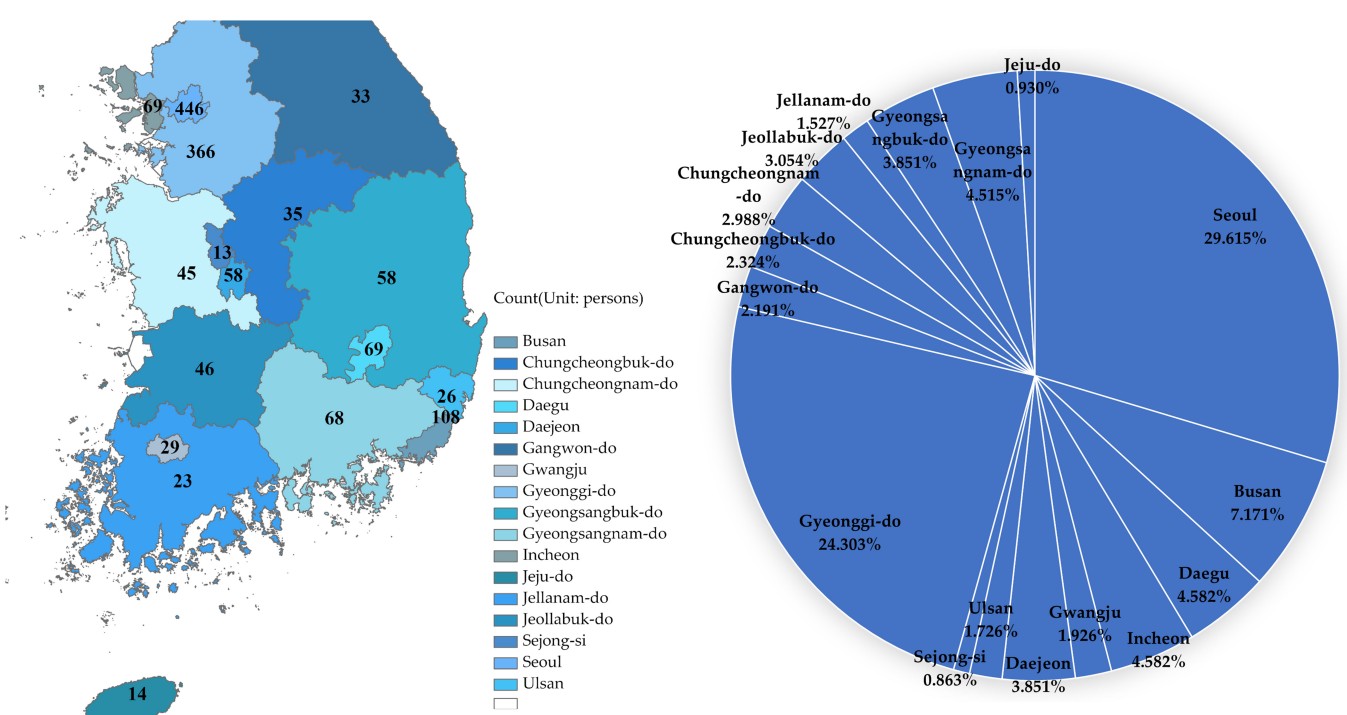

**Figure 2.** Regional distribution of the 1506 survey respondents.

By investigating the significance of the functions among the general population based on the previously mentioned maximum acceptable outage, we can assist stakeholders in their decision making regarding recovery priorities. Accordingly, a survey was conducted to ascertain the criticality (maximum acceptable outage) of the 19 national functions during a disaster. Survey results regarding a residential area served as the independent variable, whereas questions regarding the gravity of the 19 functions were used as dependent variables. The significance elicited by the survey was scored on a 5-point scale, based upon which the recovery priorities of the functions were analysed.

To determine the criticality of the 19 national functions, we incorporated the maximum acceptable outage from ISO 22301, which represents one of the main principles of disaster-recovery objectives. The term 'maximum acceptable outage' describes the maximum duration for which citizens can endure a disruption in the respective national service. This timeframe is primarily dictated by the users, that is, citizens, rather than the operational personnel, such as public officials, responsible for the national functions. Thus, when surveying the populace regarding the criticality of these 19 national functions, we designed a questionnaire using the concept of maximum acceptable outages. The criticality criteria are listed in Table 3. Criticality is categorised under levels ranging from 1 to 5, with higher levels aligning closely with 'Enormous'. As the levels decrease, the maximum allowable duration increases, potentially reaching up to one month. The standards posited for maximum acceptable outages are drawn from a Japanese guideline titled 'Guide to Business Continuity for Local Public Entities During Earthquake Occurrence and Its Commentary' [31]. This manual was compiled based on discussions by a review committee comprising staff from Japanese local government entities and related ministries, outlining the essentials for reviewing the COOP. The 'Criteria Table for Selecting Tasks by Business Resumption Objective Time' within this manual defines the maximum acceptable outage for functions disrupted by a disaster. These criteria were applied in our survey.

**Table 3.** Criteria for maximum acceptable outage.

| | Importance | Maximum Acceptable Outage |
|---|---|---|
| Stage 1 | Minor | Recovery required within one month |
| Stage 2 | Little | Recovery required within two weeks |
| Stage 3 | Medium | Recovery required within three days |
| Stage 4 | Great | Recovery required within one day |
| Stage 5 | Enormous | Recovery required within three hours |

*3.3. Analysis of the Survey Result Consistency through Cronbach's Alpha*

Cronbach's alpha was used to analyse the consistency of the survey results. Cronbach's alpha is a statistical measure frequently used in consistency analyses [42] for measuring reliability in research that is especially reliable when tau equivalence and normality are satisfied. Alternatives such as the omega coefficient can overcome the limitations of Cronbach's alpha when the assumption of tau equivalence is violated. However, GLB and GLBa show positive bias in normal distributions. In such cases, Cronbach's alpha remains a valid and appropriate choice [43]. This study used it to determine whether the 1506 respondents provided consistent responses across the 19 survey items. Cronbach's alpha was calculated using Equation (1), where $\alpha$ represents Cronbach's alpha; $N$ is the total number of items; $\sigma_i$ is the variance for each item; $\sigma_X$ is the variance of the overall score; $\sigma_i^2$ denotes the variance of a given item $i$, indicating how much the scores for that item differ; and $\sigma_X^2$ is the variance of total scores across all items, reflecting the overall score variation [44,45]. A total of 19 items were used. The variance of the overall score refers to the variance of the responses chosen by the 1506 participants for the $i$th item. In contrast, the variance of the total score represents the variance of the responses across all 19 items by the 1506 respondents. Table 4 presents the results of the analysis. The acceptability of the Cronbach's alpha was determined based on a benchmark of 0.7 [46]. Cronbach's alpha was calculated to be 0.860. It was observed that when any specific question (Q) was removed, all the values were lower than 0.860. This outcome indicates that all the items are consistent, and thus there is consistency among the responses of the 1506 adults.

$$\alpha = \left( \frac{N}{N-1} \right) * \left( 1 - \frac{\sum \sigma_i^2}{\sigma_X^2} \right) \tag{1}$$

**Table 4.** Analysis of survey data consistency using Cronbach's alpha.

| Survey No. | National Functions | | Average (N = 1506) | Standard Deviation (N = 1506) | Cronbach's Alpha (Q = 19 Items) | Cronbach's Alpha upon Removal of the Question (Q) |
|---|---|---|---|---|---|---|
| Q1 | | Recovery | 3.61 | 1.112 | | 0.853 ($\leq$0.860) |
| Q2 | Disaster Safety | Response | 4.57 | 0.754 | | 0.860 ($\leq$0.860) |
| Q3 | | Preparation | 4.08 | 1.013 | | 0.856 ($\leq$0.860) |
| Q4 | | Prevention | 3.65 | 1.083 | | 0.852 ($\leq$0.860) |
| Q5 | Public Order and Safety | | 4.21 | 0.898 | | 0.857 ($\leq$0.860) |
| Q6 | Science and Technology | | 2.61 | 1.083 | | 0.851 ($\leq$0.860) |
| Q7 | Education | | 2.70 | 1.066 | | 0.854 ($\leq$0.860) |
| Q8 | National Defence | | 4.00 | 0.998 | | 0.856 ($\leq$0.860) |
| Q9 | Agriculture, Forestry, Marine, and Fisheries | | 3.01 | 0.996 | | 0.851 ($\leq$0.860) |
| Q10 | Culture, Sports, and Tourism | | 2.40 | 1.085 | 0.860 | 0.855 ($\leq$0.860) |
| Q11 | Health Services | | 4.25 | 0.843 | | 0.855 ($\leq$0.860) |
| Q12 | Social Welfare | | 2.81 | 1.029 | | 0.850 ($\leq$0.860) |
| Q13 | Industry and SMEs | | 2.72 | 1.078 | | 0.850 ($\leq$0.860) |
| Q14 | Transportation and Logistics | | 4.09 | 0.900 | | 0.854 ($\leq$0.860) |
| Q15 | Public Administration | | 3.19 | 1.077 | | 0.850 ($\leq$0.860) |
| Q16 | Regional Development | | 2.32 | 1.092 | | 0.854 ($\leq$0.860) |
| Q17 | Telecommunications | | 4.08 | 0.929 | | 0.857 ($\leq$0.860) |
| Q18 | Unification and Diplomacy | | 3.21 | 1.081 | | 0.851 ($\leq$0.860) |
| Q19 | Environmental Protection | | 3.45 | 1.054 | | 0.852 ($\leq$0.860) |

*3.4. Analysis of the Association between Regional Differences and National Function Criticality Using Cramer's V*

After analysing the consistency of the survey data, we investigated any significant differences in the survey responses by residential area using Cramer's *V*, a method for measuring the association between categorical variables based on the chi-square test [47]. Cramer's *V* has several advantages over other statistical methodologies such as Phi, Cohen's *w*, Odds Ratio, Cohen's *h*, Freeman's Theta and Square Epsilon, Kendall's Tau-b, Goodman and Kruskal's gamma, and Somers' D when evaluating the association between nominal-level variables. Cramer's *V* can be used in cross-tabulations of all sizes and provides a clear and intuitive interpretation, with values ranging between 0 (no correlation) and 1 (perfect association). These strengths highlight Cramer's *V* as a superior choice for assessing the association between variables compared with other methodologies. Although the survey data could be precisely described as ordinal within the categorical type, the intervals between the answer values were inconsistent. Hence, we adopted Cramer's *V* to analyse simple and general correlations between categorical variables without considering ranking information. Cramer's *V* was calculated using Equation (2) [48], where *V* is a coefficient that measures the strength of the association between two categorical variables. $\chi^2$ is the chi-squared statistic for association, *n* is the total sample size, and *k* is the number of categories. The degree of correlation was analytically determined based on the following hypotheses: H0 and H1 represent the universality and non-universality of national functions, respectively.

$$V = \sqrt{\frac{\chi^2 / n}{k - 1}} \qquad (2)$$

**H0.** *The criticality of national functions does not differ by region (universal).*

**H1.** *The criticality of national functions differs by region (non-universal).*

When Cramer's *V* value was presented alongside the *p*-value, the chi-square test was performed after calculating the Cramer's *V* value to assess the independence between two categorical variables. If the *p*-value was below the significance level, H1 was adopted;

otherwise, H0 was adopted. The significance level represents the maximum allowable probability of mistakenly rejecting a true null hypothesis, that is, wrongly determining a true null hypothesis as false, which is known as a Type 1 error. In this study's context, a Type 1 error refers to the probability of mistakenly judging national functions as non-universal when they are, in fact, universal. This error can result in significant societal costs. To prevent this and enhance the reliability of the test results, the significance level was set at 0.005 [49]. Table 5 presents the results of the association analysis between the region (independent variable) and the criticality of national functions (dependent variable) using Cramer's $V$. For each function's hypothesis test result, if the $p$-value was less than 0.005, H1 was adopted; otherwise, H0 was adopted. All 19 national functions yielded a $p$-value greater than or equal to 0.005, leading to the adoption of the null hypothesis (H0), while the specific national functions showed no regional differences in Korea (universal).

**Table 5.** Results of Cramer's $V$ analysis.

| Independent Variable | Survey No. | National Functions | | Cramer's $V$ | $p$-Value | Accepted Hypothesis |
|---|---|---|---|---|---|---|
| Residential Area | Q1 | | Recovery | 0.107 | 0.319 | H0 |
| | Q2 | Disaster Safety | Response | 0.097 | 0.744 | H0 |
| | Q3 | | Preparation | 0.098 | 0.700 | H0 |
| | Q4 | | Prevention | 0.107 | 0.327 | H0 |
| | Q5 | Public Order and Safety | | 0.104 | 0.446 | H0 |
| | Q6 | Science and Technology | | 0.117 | 0.057 | H0 |
| | Q7 | Education | | 0.126 | 0.007 | H0 |
| | Q8 | National Defence | | 0.090 | 0.925 | H0 |
| | Q9 | Agriculture, Forestry, Marine, and Fisheries | | 0.099 | 0.641 | H0 |
| | Q10 | Culture, Sports, and Tourism | | 0.107 | 0.320 | H0 |
| | Q11 | Health Services | | 0.105 | 0.410 | H0 |
| | Q12 | Social Welfare | | 0.121 | 0.027 | H0 |
| | Q13 | Industry and SMEs | | 0.102 | 0.541 | H0 |
| | Q14 | Transportation and Logistics | | 0.127 | 0.005 | H0 |
| | Q15 | Public Administration | | 0.107 | 0.331 | H0 |
| | Q16 | Regional Development | | 0.100 | 0.594 | H0 |
| | Q17 | Telecommunications | | 0.111 | 0.166 | H0 |
| | Q18 | Unification and Diplomacy | | 0.114 | 0.113 | H0 |
| | Q19 | Environmental Protection | | 0.096 | 0.774 | H0 |

Although the Cramer's $V$ coefficient showed no major regional differences in public opinion, it was important to explore whether disaster-recovery priorities varied by region, considering actual resources and transportation capabilities. This involved assessing whether urban and rural areas, with their unique economic and geographical features, might have different priorities, such as urban areas focusing on health services and public order and rural areas focusing on agriculture and fisheries. Such an analysis is vital for understanding regional variations in disaster response and creating region-specific strategies, ensuring that disaster plans are grounded in both public preferences and practical resource considerations.

## 4. Establishment of Criteria for Priority of Recovering National Functions

### 4.1. Results

The consistency of the survey data was assessed using Cronbach's alpha. Additionally, the association between the criticality of national functions and location was analysed using Cramer's $V$. The survey data, validated for consistency and universality across regions, were statistically analysed to derive a ranking of the criticality of national functions. The results of the data analysis and criticality ranking are presented in Table 6. The consistency of the responses, as determined using Cronbach's alpha, was 0.860. The average result of the Cramer's $V$ analysis by region was 0.107. The primary measure of consistency based on

region, the *p*-value, exhibited an average of 0.392 and a minimum value of 0.005, indicating a significance level greater than 0.005, underscoring consistency across regions and the universality of national functions. The criticalities of the universally validated national functions were calculated using Equation (3). Here, the severity score *C* was calculated using $N_n$, the count of respondents who identified the *n*th national function as critical out of 1506 participants. The score gauged the urgency of each national function by applying weights to the responses and dividing it by the total number of 1506 respondents, thus encapsulating the overall severity for the prioritisation of national functions. The maximum value for severity was five.

$$C = 1 * \frac{N_1}{1506} + 2 * \frac{N_2}{1506} + 3 * \frac{N_3}{1506} + 4 * \frac{N_4}{1506} + 5 * \frac{N_5}{1506} \tag{3}$$

**Table 6.** Results of data analysis and criticality rankings.

| Survey No. | National Function | | Criticality Scoring (Out of 5 Points) | Consistency | Regional Cramer's *V* | Regional *p*-Value | Criticality Rank |
|---|---|---|---|---|---|---|---|
| Q1 | | Recovery | 3.6 | | 0.107 | 0.319 | 9 |
| Q2 | Disaster Safety | Response | 4.6 | | 0.097 | 0.744 | 1 |
| Q3 | | Preparation | 4.1 | | 0.098 | 0.700 | 6 |
| Q4 | | Prevention | 3.6 | | 0.107 | 0.327 | 8 |
| Q5 | Public Order and Safety | | 4.2 | | 0.104 | 0.446 | 3 |
| Q6 | Science and Technology | | 2.6 | | 0.117 | 0.057 | 17 |
| Q7 | Education | | 2.7 | | 0.126 | 0.007 | 16 |
| Q8 | National Defence | | 4.0 | | 0.090 | 0.925 | 7 |
| Q9 | Agriculture, Forestry, Marine, and Fisheries | | 3.0 | | 0.099 | 0.641 | 13 |
| Q10 | Culture, Sports, and Tourism | | 2.4 | 0.860 | 0.107 | 0.320 | 18 |
| Q11 | Health Services | | 4.3 | | 0.105 | 0.410 | 2 |
| Q12 | Social Welfare | | 2.8 | | 0.121 | 0.027 | 14 |
| Q13 | Industry and SMEs | | 2.7 | | 0.102 | 0.541 | 15 |
| Q14 | Transportation and Logistics | | 4.1 | | 0.127 | 0.005 | 4 |
| Q15 | Public Administration | | 3.2 | | 0.107 | 0.331 | 12 |
| Q16 | Regional Development | | 2.3 | | 0.100 | 0.594 | 19 |
| Q17 | Telecommunications | | 4.1 | | 0.111 | 0.166 | 5 |
| Q18 | Unification and Diplomacy | | 3.2 | | 0.114 | 0.113 | 11 |
| Q19 | Environmental Protection | | 3.5 | | 0.096 | 0.774 | 10 |
| | Average | | 3.4 | 0.860 | 0.107 | 0.392 | - |

Figure 3 presents the criticality of national functions in terms of rankings. The top three functions exhibiting the highest criticality are Disaster Safety Response, Health Services, and Public Order and Safety. These functions serve as the fundamental pillars of a nation and play pivotal roles in safeguarding citizens' lives and safety. The significance of these functions has increased recently, particularly in the wake of various societal events and disaster scenarios. Such observations underscore the public's perception of these functions as paramount among national functions.

The functions with the lowest perceived criticality were Science and Technology and Regional Development. Although these sectors contribute significantly to enhancing the quality of life of citizens and their overall progress, their lack of direct implications for immediate life and safety may have contributed to their lower criticality ratings. It is imperative to understand that these findings do not trivialise or underplay the significance of these sectors. Instead, they merely reflect the public's relative priorities and concerns. These results underscore the state's need to establish and implement disaster-response policies that resonate with public priorities and needs. The high priority assigned to environmental protection in this study can be interpreted as a reflection of public priorities and concerns. This emphasises the importance of public opinion and needs in setting recovery priorities for national functions in the event of a disaster. Furthermore, these

insights can aid in the efficient allocation of resources in public service delivery based on criticality and assist in policy formulation and budget allocation for each function.

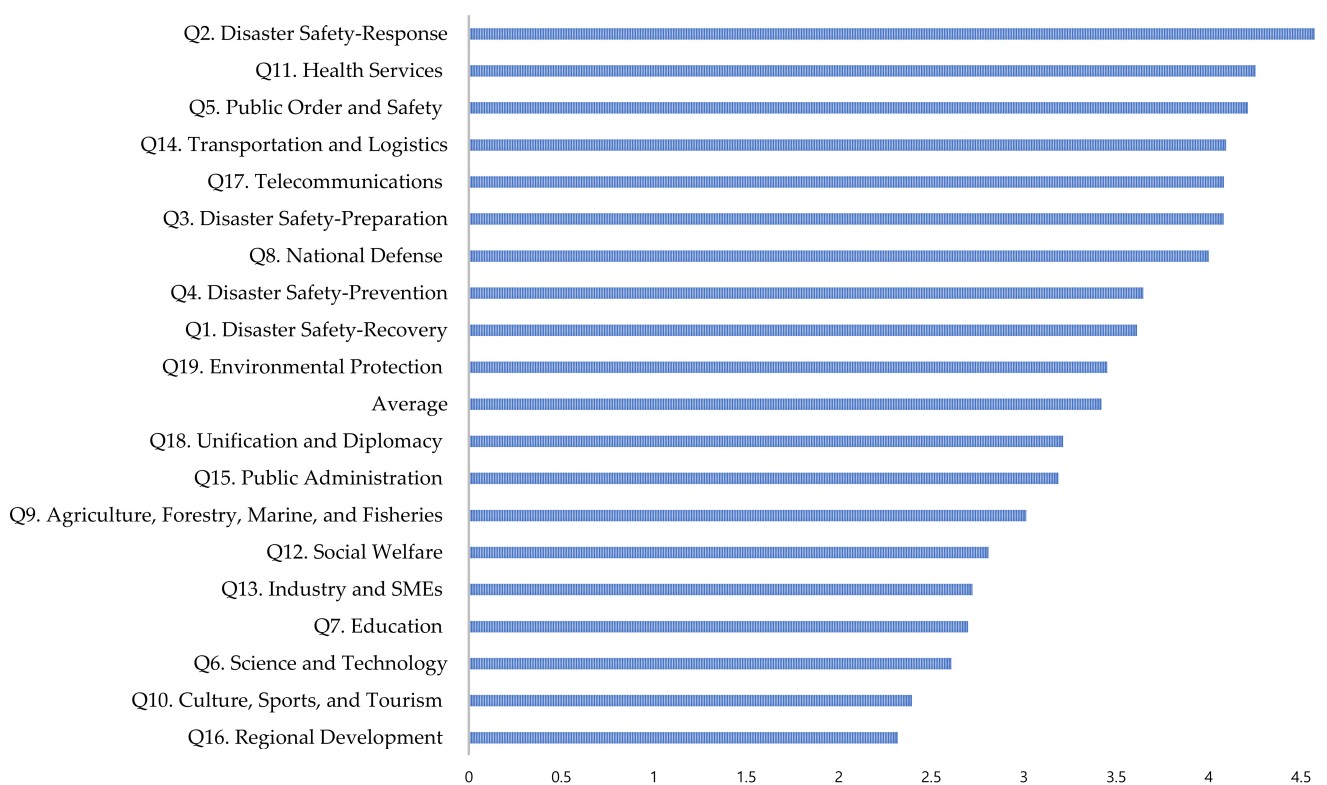

**Figure 3.** Ranking results of national function criticality.

## 4.2. Potential Implications of National Function Priorities in Emergency Situations

The potential impacts of national functional priorities in emergency situations can be summarised as follows. First, based on data collected before the COVID-19 pandemic, we analysed the significance of national functions in general emergency situations. Such analyses can provide essential insights for government decision making in unprecedented global crises such as pandemics. This will provide clear guidelines regarding which functions must be urgently restored in the event of a disaster. Moreover, these priority settings provide essential foundational data for formulating disaster-response policies and budget allocations. This contributes to the establishment of a basis for efficient resource allocation and policy development in disaster situations.

Second, this study examines the balance between public health and the economy in adapting general emergency-situation priorities to pandemic contexts, highlighting their crucial role. It emphasises the government's significant responsibility in balancing public health, which is directly linked to citizens' lives, with economic stability, which is crucial for quality of life. The findings offer insights for prioritising national functions during pandemics and can aid governments in understanding the challenges of current and future global crises. This complex task involves not only policy decisions, but also the simultaneous consideration of public health and economic stability.

Third, this study lays a crucial foundation for setting disaster-recovery priorities for national functions. The key findings suggest prioritising disaster safety responses, health services, and public order during crises, indicating that governments should focus on public safety and health amidst conflicting interests. This study also highlights the necessity for further analyses to specify recovery priorities for different disaster scenarios, which will help in efficient resource allocation and policy formulation in line with public needs during such events.

Fourth, incorporating average levels in the priority ranking of post-disaster recovery enables a stratified approach to address priority factors effectively. This stratification distinguishes between immediate critical needs (first level), secondary level guarantees, and subsequent reconstruction efforts, ensuring a more organised and efficient response to disaster scenarios.

## 5. Conclusions

In this study, we derived 19 national functions from Republic of Korea's Business Reference Model and conducted a survey analysis of the criticality of each function among 1506 citizens (including 123 public officials). The frequency of the survey responses was analysed to determine the criticality of each function. Statistical analyses were employed to determine the consistency (Cronbach's alpha) and correlation (Cramer's *V*) of the criticality responses across regions. The calculated value of Cronbach's alpha was 0.860 and the average Cramer's *V* was 0.107. The average *p*-value, as determined by Cramer's *V* test, was 0.0392. The null hypothesis (H0), 'There is no regional difference in the criticality of national functions', was adopted, verifying the universality of the national functions. We derived recovery priorities for national functions based on the validated criticality scores. The Disaster Safety-Response function had the highest priority, and the Regional Development function ranked the lowest. This study categorises national functions and analyses their recovery priorities based on criticality, overcoming the limitations of previous research. Cronbach's alpha and Cramer's *V* demonstrate consistent and universally important responses across regions, which are crucial for efficient recovery prioritisation in disaster management. This study offers a foundational model for setting restoration priorities post-disaster that is applicable in diverse contexts.

This study highlights the effects of frequent and severe disasters on various national functions. Unlike previous research, which has partially addressed these functions in disaster scenarios, this study provides a thorough analysis and sets recovery priorities using a South Korean Business Reference Model. This innovative approach for classifying national functions and recovery strategies is expected to improve resource efficiency and societal resilience during disasters. The results of this study provide a foundation for establishing recovery priorities in the event of disaster-induced interruptions in national functions. It is anticipated that, during disaster-triggered interruptions, restoring the highest-priority functions, such as Disaster Safety Response, Health Services, and Public Order and Safety, will pre-emptively mitigate public discontent. Furthermore, we expect that advanced nations, including the United States, Japan, and the EU countries, can reference these recovery priority standards during disaster scenarios.

However, this study has several limitations. First, while the study sets recovery priorities for national functions based on the public will during disasters, we recognise the need for more discussion on government resource limitations. This study should include a detailed methodology for resource prioritisation. This will help in evaluating the practicality of government resource allocation and policymaking, while assessing the importance of each national function. This approach is vital for aligning public priorities with government resource constraints. Second, despite validating the recovery priorities through survey analysis, this study has a limitation—the functions are not detailed. Further analytical research on recovery priorities at the broader functional level is essential to determine recovery priorities for subdivisions, such as medical service support and emergency medical system operations within the health function. Additionally, while the study presented findings on the international universality of functions in Section 3, a review of functions from other countries and international surveys is required to further generalise the recovery priority results. Third, we plan to compare the pre-pandemic data from this study with post-pandemic data in future research. This comparison will reveal how national function priorities shifted because of the pandemic and help identify new disaster management trends and challenges post-pandemic, thereby enhancing our understanding of effective response strategies in an evolving global context. Moreover, in addition to global pandemics,

it is necessary to consider scenarios such as earthquakes, tsunamis, and states of war. Lastly, future studies should include not only diverse populations from different regions and public officials but also consider additional demographic factors such as occupation, income, and education level to gain more detailed insights into how societal segments perceive the importance of national functions, which may vary according to socioeconomic backgrounds. In addition to demographic factors, a statistical analysis of the occurrence rankings of previous disasters should also be included. Nevertheless, the outcomes of this study can serve as foundational data for determining the recovery priorities for interrupted national functions following disasters.

**Author Contributions:** Conceptualisation, H.S.Y.; methodology, S.B.H.; software, J.B.L.; validation, J.H.S.; formal analysis, J.B.L.; investigation, S.B.H.; resources, J.H.S.; data curation, J.B.L.; writing—original draft preparation, S.B.H.; writing—review and editing, S.B.H.; visualisation, J.H.S.; supervision, H.S.Y.; project administration, H.S.Y.; funding acquisition, H.S.Y. All authors have read and agreed to the published version of the manuscript.

**Funding:** This research was supported by the 2023-MOIS36-004(RS-2023-00248092) of the Technology Development Program on Disaster Restoration Capacity Building and Strengthening funded by the Ministry of Interior and Safety (MOIS, Republic of Korea).

**Institutional Review Board Statement:** Not applicable.

**Informed Consent Statement:** Not applicable.

**Data Availability Statement:** Data are available upon request due to restrictions (e.g., privacy or ethical).

**Conflicts of Interest:** The authors declare no conflict of interest.

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
