# Peer review of "Criteria for and Policy Implications of Setting Recovery Priorities of National Functions during Disruptions by Disasters"

_sustainability, doi:10.3390/su152416615_

Round 1
Reviewer 1 Report
Comments and Suggestions for Authors
In this paper, 19 National functions were extracted from the Korea Integrated Business Reference Model and citizens and government were surveyed officials to determine the importance of each function. Statistical analyses and confirmed the consistency and relevance of the answers from different regions. Based on the validated importance values, national functions were prioritised for recovery, with "disaster safety response" being prioritised as the highest function and "regional development" being ranked the lowest. The results of the study are instructive, but some issues remain. Specific issues are set out below:
1、Are the formulas in the article original formulas? If not please cite them.
2、Please provide an explanation of each parameter in the formulas. Formula (2), for example, should be explained below it: where x represented the ......, n the ......, k the ......
3、How can authors verify the reliability of the chosen model? It is suggested that the authors can do a comparative analysis of different models as a way to increase the persuasiveness of the chosen model.
4、It is recommended to summarize the innovations of this article in the conclusion section.
5、The article's statements lack clarity and require further condensation, and the level of innovation must be improved.
Comments on the Quality of English Languagepoor
Author Response
<Reviewer #1>
In this paper, 19 National functions were extracted from the Korea Integrated Business Reference Model and citizens and government were surveyed officials to determine the importance of each function. Statistical analyses and confirmed the consistency and relevance of the answers from different regions. Based on the validated importance values, national functions were prioritised for recovery, with "disaster safety response" being prioritised as the highest function and "regional development" being ranked the lowest. The results of the study are instructive, but some issues remain. Specific issues are set out below:
1.Are the formulas in the article original formulas? If not please cite them.
Our response: Thank you for your good comment. I would like to add more detailed references for the formulas, as further elaboration is necessary. I have added references for Equation (1) and (2) as shown below, and Equation (3) is a formula I created myself.
(line 585) Wikipedia. Cronbach's alpha. Available online: https://en.wikipedia.org/wiki/Cronbach%27s_alpha (accessed on 2023.11.23).
(line 589) Wikipedia. Cramér's V. Available online: https://en.wikipedia.org/wiki/Cram%C3%A9r%27s_V (accessed on 2023.11.23).
- 2. Please provide an explanation of each parameter in the formulas. Formula (2), for example, should be explained below it: where x represented the ......, n the ......, k the ......
Our response: Thank you for your good comment. I have added explanations for each parameter in equations (1) to (3) as follows.
(line 273) Cronbach’s alpha was calculated using Equation (1), where α represents Cronbach’s alpha, N is the total number of items, is the variance for each item, is the variance of the overall score, denotes the variance of a given item i, indicating how much the scores for that item differ, and is the variance of total scores across all items, reflecting the overall score variation. [
(line 312) Hence, we adopted Cramer’s V to analyse simple and general correlations between categorical variables without considering ranking information. Cramer’s V was calculated using Equation (2), where is a coefficient that measures the strength of the association between two categorical variables. is the chi-squared statistic for association, is the total sample size, and is the number of categories.
(line 362) The criticalities of the universally validated national functions were calculated using Equation (3). Here, the severity score C was calculated using Nn, the count of respondents who identified the nth national function as critical out of 1,506 participants. The score gauged the urgency of each national function by applying weights to the responses and dividing it by the total number of 1,506 respondents, thus encapsulating the overall severity for the prioritisation of national functions.
- How can authors verify the reliability of the chosen model? It is suggested that the authors can do a comparative analysis of different models as a way to increase the persuasiveness of the chosen model.
Our response: Thank you for your insight comment. My research is based on a recovery priority model derived from the Business Reference Model, and the statistical analysis methodology employed includes Cronbach’s alpha analysis and Cramer’s V analysis.
In the paper, the Table 2(line 215) refers to similar national functions compared to the Business Reference Model (BRM). Additionally, I have added content on the summary and comparative analysis of the Business Reference Model and the priority model.
(line 212) This study conducted a comparative analysis between South Korea’s Business Reference Model and the national function classification systems of the United States, Japan, and the EU. The U.S. ‘National Critical Functions Set’ matched all functions of the South Korean model, while Japan and the EU’s models showed similar structures. This international comparison underscores the high universality and excellence of South Korea’s Business Reference Model in the national function classification.
The content on the summary and comparative analysis of methodologies similar to Cronbach’s alpha analysis has been added as follows.
(line 273) Cronbach’s alpha was used to analyse the consistency of the survey results. Cronbach’s alpha is a statistical measure frequently used in consistency analyses for measuring reliability in research that is especially reliable when tau equivalence and normality are satisfied. Alternatives such as the omega coefficient can overcome the limitations of Cronbach’s alpha when the assumption of tau equivalence is violated. However, GLB and GLBa show positive bias in normal distributions. In such cases, Cronbach’s alpha remains a valid and appropriate choice.
The content regarding the summary and comparative analysis of methodologies similar to Cramer’s V analysis has been added as follows.
(line 304) Cramer’s V has several advantages over other statistical methodologies such as Phi, Cohen’s w, Odds Ratio, Cohen’s h, Freeman’s Theta and Square Epsilon, Kendall’s Tau-b, Goodman and Kruskal’s gamma, and Somers’ D when evaluating the association between nominal-level variables. Cramer’s V can be used in cross-tabulations of all sizes and provides a clear and intuitive interpretation, with values ranging between 0 (no correlation) and 1 (perfect association). These strengths highlight Cramer’s V as a superior choice for assessing the association between variables compared with other methodologies.
- It is recommended to summarize the innovations of this article in the conclusion section.
Our response: Thank you for your good comment. In the conclusion section, I summarized the innovation of the article as follows. To avoid overlap with the content in reviewer's comment 5, the study addressed innovation in terms of statistical validation.
(line 437) This study categorises national functions and analyses their recovery priorities based on criticality, overcoming the limitations of previous research. Cronbach’s alpha and Cramer’s V demonstrate consistent and universally important responses across regions, which are crucial for efficient recovery prioritisation in disaster management. This study offers a foundational model for setting restoration priorities post-disaster that is applicable in diverse contexts.
- The article's statements lack clarity and require further condensation, and the level of innovation must be improved.
Our response: Thank you for this good comment. It is necessary to summarize the background and distinctiveness of the paper and to make additional remarks on the level of innovation.
Further elaboration on the level of innovation is provided as follows.
(line 443) This study highlights the effects of frequent and severe disasters on various national functions. Unlike previous research, which has partially addressed these functions in disaster scenarios, this study provides a thorough analysis and sets recovery priorities using a South Korean Business Reference Model. This innovative approach for classifying national functions and recovery strategies is expected to improve resource efficiency and societal resilience during disasters.
The background and distinctiveness of the paper are mentioned as follows.
(line 125) Previous research has only partially addressed national functions in disaster scenarios. This study extends this by comprehensively analysing all national functions using the South Korean Business Reference Model with a focus on their classification and recovery prioritisation. A nationwide survey assessed the criticality of national functions, yielding a recovery priority model for efficient resource utilisation during disasters. This marks a significant innovation in national function classifications and recovery strategies.

Reviewer 2 Report
Comments and Suggestions for Authors
1. What is the government's priority in times of global crisis?
2. In the Covid-19 pandemic, was the government's priority the people's health or the country's economy?
3. In conflicting interests, what is the government's priority?
Comments on the Quality of English LanguageMinor editing of English language required
Author Response
<Reviewer #2>
- What is the government's priority in times of global crisis?
Our response: Thank you for your good comment. In my research, I focused on how governments prioritize the recovery of national functions in national crisis situations, particularly during disasters. This concept, as mentioned in the Table 2(lines 215), can similarly be applied to other advanced nations such as the United States, Japan, and the EU. It was discovered that the 'Disaster Safety-Response' function is assigned the highest recovery priority in crisis situations. Conversely, other functions like 'Regional Development' are assigned a relatively lower priority. Regarding global crises, I have added the following text to the paper.
(line 125) Previous research has only partially addressed national functions in disaster scenarios. This study extends this by comprehensively analysing all national functions using the South Korean Business Reference Model with a focus on their classification and recovery prioritisation. A nationwide survey assessed the criticality of national functions, yielding a recovery priority model for efficient resource utilisation during disasters. This marks a significant innovation in national function classifications and recovery strategies.
- In the Covid-19 pandemic, was the government's priority the people's health or the country's economy?
Our response: Thank you for providing insightful foresight. The scope of this research is set before the pandemic, and specific priorities during the COVID-19 pandemic were not directly addressed. However, the findings of this research are considered universally important for national functions in general emergency situations. Cronbach's alpha was calculated at 0.860 and Cramer’s V averaged 0.107. The discussion below has been added on how the national function priorities could be transformed in a pandemic scenario when applying the pandemic as an emergency situation.
(line 396)
The potential impacts of national functional priorities in emergency situations can be summarised as follows. First, based on data collected before the COVID-19 pandemic, we analysed the significance of national functions in general emergency situations. Such analyses can provide essential insights for government decision-making in unprecedented global crises such as pandemics. This will provide clear guidelines regarding which functions must be urgently restored in the event of a disaster. Moreover, these priority settings provide essential foundational data for formulating disaster response policies and budget allocations. This contributes to the establishment of a basis for efficient resource allocation and policy development in disaster situations.
Second, this study examines the balance between public health and the economy in adapting general emergency situation priorities to pandemic contexts, highlighting their crucial role. It emphasises the government’s significant responsibility in balancing public health, which is directly linked to citizens’ lives, with economic stability, which is crucial for quality of life. The findings offer insights for prioritising national functions during pandemics and can aid governments in understanding the challenges of current and future global crises. This complex task involves not only policy decisions, but also the simultaneous consideration of public health and economic stability.
- In conflicting interests, what is the government's priority?
Our response: Thank you for providing the opportunity to further discuss this research. The survey conducted in this study assessed the criticality (maximum allowable interruption time) of 19 national functions in the event of a disaster, and based on these results, determined recovery priorities. This research included consistency analysis using Cronbach's alpha and correlation analysis of location-based national function importance using Cramer’s V. This emphasizes the consistency across regions and the universality of national functions. The findings of this study lay the groundwork for establishing recovery priorities in the event of a national function disruption due to disasters. It is expected that restoring highest priority functions such as disaster safety response, health services, public order, and safety in the event of a disruption will help preemptively mitigate public dissatisfaction. While this study verified recovery priorities through survey analysis, it did not address specific details of individual functions, and further analytical research on broader function-level recovery priorities is necessary. The following content has been added to address this.
(line 413) Third, this study lays a crucial foundation for setting disaster recovery priorities for national functions. The key findings suggest prioritising disaster safety responses, health services, and public order during crises, indicating that governments should focus on public safety and health amidst conflicting interests. This study also highlights the necessity for further analyses to specify recovery priorities for different disaster scenarios, which will help in efficient resource allocation and policy formulation in line with public needs during such events.

Reviewer 3 Report
Comments and Suggestions for Authors
Based on the Public Records Management Act of the Korean Government, 19 indicators related to disaster response at the level of policy departments are selected to investigate the "maximum acceptable interruption" of different indicators. Based on this, the paper tests the universality of using score weighting to determine the priority of disaster response in different regions, which is novel and practical significance. Good topic selection; The overall research idea is clear, and the conclusion is valuable for reference. Review conclusion: It is recommended to be revised and approved for publication. Some suggestions to be further improved:
(1) Several key scientific issues need to be further sorted out and improved: First, in the priority ranking, environmental protection is higher than public management and social services, which ordinary people pay more attention to. Is there any special factors and social background? Second, if the average level is included in the priority ranking, can we stratified the corresponding priority factors after the disaster according to it, and distinguish the first level, namely the first guarantee problem, the second level guarantee and the subsequent reconstruction work levels? Third, in the selection process of 19 index elements, the policies of several major countries were compared and combined, and the corresponding sequence was not combined with comprehensive analysis of other countries. Is it a priority of post-disaster response only applicable to the national conditions of South Korea? The post-disaster response sequence is basically in line with Maslow's demand theory, and is its universality high?
(2) There is a slight lack of research priority order, which is not well related to the previous article; This paper considers the priority issue based on the limited government resources, but the actual method only considers the public's willingness to investigate, and whether the resource limitation part should be discussed in addition.
(3) The comprehensive analysis of whether the priority order is applicable to each region needs to be strengthened. The cramer v coefficient is used in the paper to show that there is no regional difference in the survey of people's will, but whether the priority order is slightly different in the context of the actual resource supply and transportation in different regions in the case of disasters can be discussed.
(4) Suggestions regarding the priority of factors in the research results can be slightly supplemented.
(5) The standardization and wording of references need to be strengthened.
(6) It is suggested to reorganize the abstract and conclusion, as well as the title.
Comments on the Quality of English Languagegood
Author Response
<Reviewer #3>
Based on the Public Records Management Act of the Korean Government, 19 indicators related to disaster response at the level of policy departments are selected to investigate the "maximum acceptable interruption" of different indicators. Based on this, the paper tests the universality of using score weighting to determine the priority of disaster response in different regions, which is novel and practical significance. Good topic selection; The overall research idea is clear, and the conclusion is valuable for reference. Review conclusion: It is recommended to be revised and approved for publication. Some suggestions to be further improved:
(1-1) Several key scientific issues need to be further sorted out and improved: First, in the priority ranking, environmental protection is higher than public management and social services, which ordinary people pay more attention to. Is there any special factors and social background?
Our response: This study provides a significant foundation for establishing recovery priorities of national functions in the event of a disaster. According to the research findings, the national functions that should be restored as top priorities during disasters include disaster safety response, health services, public order, and safety. These results suggest that in situations of conflicting interests, governments should prioritize public safety and health when deciding on recovery priorities. Moreover, this research emphasizes the need for additional analytical studies to determine the recovery priorities of specific functions in certain disaster situations. This is expected to assist governments in effectively allocating resources and formulating policies that align with public demands and priorities during disaster situations. The information added to the text regarding this is as follows.
(line 387) The high priority assigned to environmental protection in this study can be interpreted as a reflection of public priorities and concerns. This emphasises the importance of public opinion and needs in setting recovery priorities for national functions in the event of a disaster. Furthermore, these insights can aid in the efficient allocation of resources in public service delivery based on criticality and assist in policy formulation and budget allocation for each function.
(1-2) Second, if the average level is included in the priority ranking, can we stratified the corresponding priority factors after the disaster according to it, and distinguish the first level, namely the first guarantee problem, the second level guarantee and the subsequent reconstruction work levels?
After a disaster, including average levels in the priority ranking allows for an effective stratification of the corresponding priority factors. This stratification helps in distinguishing between immediate and critical needs (first level), secondary level guarantees, and reconstruction work, enabling a more organized and efficient response to disaster situations. The following text has been added to the document to reflect this approach.
(line 420) Fourth, incorporating average levels in the priority ranking of post-disaster recovery enables a stratified approach to address priority factors effectively. This stratification distinguishes between immediate critical needs (first level), secondary level guarantees, and subsequent reconstruction efforts, ensuring a more organized and efficient response to disaster scenarios.
(1-3) Third, in the selection process of 19 index elements, the policies of several major countries were compared and combined, and the corresponding sequence was not combined with comprehensive analysis of other countries. Is it a priority of post-disaster response only applicable to the national conditions of South Korea? The post-disaster response sequence is basically in line with Maslow's demand theory, and is its universality high?
Based on the research presented in this study, the method of setting post-disaster response priorities, while grounded in the context of South Korea and aligned with Maslow's theory, also incorporates a comparison of government functions with several major countries (Table 2, line ~~). The 19 index elements derived from this comparison align with fundamental human needs, suggesting that this methodology has a high potential for universal application beyond South Korea. The following text has been added to the document to reflect this aspect.
(line 192) Based on the research presented in this study, the approach to setting post-disaster response priorities is primarily grounded in the national context of South Korea and Maslow's hierarchy of needs, yet it exhibits a high degree of universality. The 19 index elements derived from comparing policies of several major countries broadly align with fundamental human needs, suggesting their applicability beyond the specific national conditions of South Korea.
(2) There is a slight lack of research priority order, which is not well related to the previous article; This paper considers the priority issue based on the limited government resources, but the actual method only considers the public's willingness to investigate, and whether the resource limitation part should be discussed in addition.
Our response: As mentioned by the reviewer, while the study focused on determining priorities based on public will, it may be important to include further discussion regarding the limitations of government resources. To address this, the methodology for determining priorities considering the limitations of government resources could be discussed in more detail. This approach could provide guidance for efficient resource allocation and policy formulation based on the insights gained. The following text has been added to address this.
(line 455) However, this study has several limitations. First, while the study sets recovery priorities for national functions based on the public will during disasters, we recognise the need for more discussion on government resource limitations. This study should include a detailed methodology for resource prioritisation. This will help in evaluating the practicality of government resource allocation and policymaking, while assessing the importance of each national function. This approach is vital for aligning public priorities with government resource constraints.
(3) The comprehensive analysis of whether the priority order is applicable to each region needs to be strengthened. The cramer v coefficient is used in the paper to show that there is no regional difference in the survey of people's will, but whether the priority order is slightly different in the context of the actual resource supply and transportation in different regions in the case of disasters can be discussed.
Our response: Thank you for you good comment. As the reviewer pointed out, a discussion on how regional differences in actual resource supply and transportation impact priority decisions in disaster situations may be necessary. This suggests the importance of considering regional characteristics and needs when establishing disaster response plans. The following text has been added to address this.
(line 338) Although the Cramer’s V coefficient showed no major regional differences in public opinion, it was important to explore whether disaster recovery priorities varied by region, considering actual resources and transportation capabilities. This involved assessing whether urban and rural areas, with their unique economic and geographical features, might have different priorities, such as urban areas focusing on health services and public order and rural areas focusing on agriculture and fisheries. Such an analysis is vital for understanding regional variations in disaster response and creating region-specific strategies, ensuring that disaster plans are grounded in both public preferences and practical resource considerations.
(4) Suggestions regarding the priority of factors in the research results can be slightly supplemented.
Our response: Thank you for your insight comment. Considering the study's findings, this paper adds further suggestions on the priorities of factors identified. The research determined the recovery priorities of national functions through a survey, employing Cronbach's alpha and Cramer's V analyses to assess consistency and regional differences. Additional suggestions on utilizing these findings have been added as follows.
(line 396) The potential impacts of national functional priorities in emergency situations can be summarised as follows. First, based on data collected before the COVID-19 pandemic, we analysed the significance of national functions in general emergency situations. Such analyses can provide essential insights for government decision-making in unprecedented global crises such as pandemics. This will provide clear guidelines regarding which functions must be urgently restored in the event of a disaster. Moreover, these priority settings provide essential foundational data for formulating disaster response policies and budget allocations. This contributes to the establishment of a basis for efficient resource allocation and policy development in disaster situations.
(5) The standardization and wording of references need to be strengthened.
Our response: Thank you for good comment. The reference section's 'The standardization and wording' has been strengthened as follows. We have added the number of pages, report numbers, and access dates for references with some information missing.
- Japan, C.O.o. Guide to Business Continuity for Local Public Entities During Earthquake Occurrence and Its Commentary, 1st Edition; 2010; p. 24~36. (line 556)
- Wong, W.N.Z.Z.; Shi, J. Business continuity management system: A complete guide to implementing ISO 22301; Kogan Page Publishers: 2014; pp. 1-273. (line 564)
- Yun, H.S. Final Report on the Development of a Disaster Risk Reduction Activity Management System Support Program and Management Technology; MPSS-Nature-2015-80; Ministry of the Interior and Safety: Seoul, 2018; p. 931. (line 568)
- CISA. National Critical Functions Set. Available online: https://www.cisa.gov/national-critical-functions-set (accessed on 2023.10.21). (line 570)
- Juseong, H.; Wontae, L.; Sunhee, C.; Dongi, S.; Woogi, L. Research on Architecture Policy for Knowledge Informatization - A Study on the Development Strategy of a Government Integrated Computer Center Based on EA (Enterprise Architecture); 9788982424205; Gwacheon: Information and Communication Policy Institute: 2008; p. 188. (line 572)
- Gøtze, J. European Interoperability Reference Architecture. Available online: https://eavoices.com/2015/07/24/european-interoperability-reference-architecture/ (accessed on 2023.10.21). (line 575)
(6) It is suggested to reorganize the abstract and conclusion, as well as the title.
Our response: Thank you for good comment. Reflecting the reviewer's comments, the title and abstract have been revised as follows. The abstract has been modified in compliance with the 200-character limit. In the conclusion section, “future plans” and “innovativeness of research, etc.” were added.
Revised title:
(Before change) Criteria for Establishing Recovery Priorities for National Functions During Disruptions Caused by Disaster
(After change) Criteria for and Policy Implications of Setting Recovery Priorities of National Functions during Disruptions by Disasters
Revised abstract: National functions are categories of operations prioritised for restoration when disrupted by emergencies such as disasters. However, the simultaneous restoration of all national functions when some or all are paralysed is limited by time and resources. Delays in the restoration of key functions can lead to public dissatisfaction. Thus, it is necessary to broadly classify national functions and analyse their restoration priorities based on criticality. This study identifies 19 national functions from South Korea’s comprehensive Business Reference Model. A survey was conducted among citizens and officials to determine the criticality of each function. Statistical analyses verified the consistency (Cronbach’s alpha = 0.860) and correlation (average Cramer’s V = 0.107) of the criticality responses across regions. The null hypothesis of no regional differences in the criticality of national functions was accepted, validating their universality. Restoration priorities were derived from these criticality values, with ‘Disaster Safety-Response’ as the highest priority and ‘Regional Development’ as the lowest. These results provide foundational data for the post-disaster restoration priorities of national functions and emphasise the need to consider public opinion, needs, and government resource limitations in disaster management planning.
Revised conclusions:
In this study, we derived 19 national functions from South Korea’s Business Reference Model and conducted a survey analysis of the criticality of each function among 1,506 citizens (including 123 public officials). The frequency of the survey responses was analysed to determine the criticality of each function. Statistical analyses were employed to determine the consistency (Cronbach’s alpha) and correlation (Cramer’s V) of the criticality responses across regions. The calculated value of Cronbach’s alpha was 0.860 and the average Cramer’s V was 0.107. The average p-value, as determined by Cramer’s V test, was 0.0392. The null hypothesis (H0), ‘There is no regional difference in the criticality of national functions’, was adopted, verifying the universality of the national functions. We derived recovery priorities for national functions based on the validated criticality scores. The Disaster Safety-Response function had the highest priority, and the Regional Development function ranked the lowest. This study categorises national functions and analyses their recovery priorities based on criticality, overcoming the limitations of previous research. Cronbach’s alpha and Cramer’s V demonstrate consistent and universally important responses across regions, which are crucial for efficient recovery prioritisation in disaster management. This study offers a foundational model for setting restoration priorities post-disaster that is applicable in diverse contexts.
This study highlights the effects of frequent and severe disasters on various national functions. Unlike previous research, which has partially addressed these functions in disaster scenarios, this study provides a thorough analysis and sets recovery priorities using a South Korean Business Reference Model. This innovative approach for classifying national functions and recovery strategies is expected to improve resource efficiency and societal resilience during disasters. The results of this study provide a foundation for establishing recovery priorities in the event of disaster-induced interruptions in national functions. It is anticipated that, during disaster-triggered interruptions, restoring the highest-priority functions, such as Disaster Safety-Response, Health Services, and Public Order and Safety, will pre-emptively mitigate public discontent. Furthermore, we expect that advanced nations, including the United States, Japan, and the EU countries, can reference these recovery priority standards during disaster scenarios.
However, this study has several limitations. First, while the study sets recovery priorities for national functions based on the public will during disasters, we recognise the need for more discussion on government resource limitations. This study should include a detailed methodology for resource prioritisation. This will help in evaluating the practicality of government resource allocation and policymaking, while assessing the importance of each national function. This approach is vital for aligning public priorities with government resource constraints. Second, despite validating the recovery priorities through survey analysis, this study has a limitation—the functions are not detailed. Further analytical research on recovery priorities at the broader functional level is essential to determine recovery priorities for subdivisions, such as medical service support and emergency medical system operations within the health function. Additionally, while the study presented findings on the international universality of functions in Section 3, a review of functions from other countries and international surveys is required to further generalise the recovery priority results. Third, we plan to compare the pre-pandemic data from this study with post-pandemic data in future research. This comparison will reveal how national function priorities shifted because of the pandemic and help identify new disaster management trends and challenges post-pandemic, thereby enhancing our understanding of effective response strategies in an evolving global context. Moreover, in addition to global pandemics, it is necessary to consider scenarios such as earthquakes, tsunamis, and states of war. Lastly, future studies should include not only diverse populations from different regions and public officials but also consider additional demographic factors such as occupation, income, and education level to gain more detailed insights into how societal segments perceive the importance of national functions, which may vary according to socioeconomic background. In addition to demographic factors, a statistical analysis of the occurrence rankings of previous disasters should also be included. Nevertheless, the outcomes of this study can serve as foundational data for determining the recovery priorities for interrupted national functions following disasters.

Reviewer 4 Report
Comments and Suggestions for Authors
This paper is recommended for acceptance, but it needs to go through some modifications, which are as follows:
1、According to the requirements of the journal to modify the reference format, such as 35, 37-40.
2、The format of formula 2 (line 303) and formula 3 (line 341) in the text should be improved.
3、From section 3.2, it is known that the main source of data for the authors' study is 2018. The global epidemic was used as the background of the study, so should the data after the global epidemic be studied again.
4、The population of the survey study came from different regions of the author's country and whether they were public servants or not, so should other conditions such as occupation, income, and education level of the survey population be considered.
5、Should V in section 3.4 be in italics.
6、The text size and style in Figures 2 and 3 in the text should be uniform.
7、The font size of the formulas in the text is not consistent.
8、The authors mainly rank state function recovery and do not incorporate the rank of disaster occurrence. The ranking of disaster occurrence and the ranking of state function recovery are also different.
9、The research result is a research study done before the global epidemic, and whether its findings are verified during the epidemic.
10、The authors' study of disasters focuses on epidemics, whether they consider the order of restoration of state functions in larger natural disasters such as earthquakes, tsunamis or states of war.
Comments on the Quality of English LanguageThis paper is recommended for acceptance, but it needs to go through some modifications, which are as follows:
1、According to the requirements of the journal to modify the reference format, such as 35, 37-40.
2、The format of formula 2 (line 303) and formula 3 (line 341) in the text should be improved.
3、From section 3.2, it is known that the main source of data for the authors' study is 2018. The global epidemic was used as the background of the study, so should the data after the global epidemic be studied again.
4、The population of the survey study came from different regions of the author's country and whether they were public servants or not, so should other conditions such as occupation, income, and education level of the survey population be considered.
5、Should V in section 3.4 be in italics.
6、The text size and style in Figures 2 and 3 in the text should be uniform.
7、The font size of the formulas in the text is not consistent.
8、The authors mainly rank state function recovery and do not incorporate the rank of disaster occurrence. The ranking of disaster occurrence and the ranking of state function recovery are also different.
9、The research result is a research study done before the global epidemic, and whether its findings are verified during the epidemic.
10、The authors' study of disasters focuses on epidemics, whether they consider the order of restoration of state functions in larger natural disasters such as earthquakes, tsunamis or states of war.
Author Response
<Reviewer #4>
This paper is recommended for acceptance, but it needs to go through some modifications, which are as follows:
- 1. According to the requirements of the journal to modify the reference format, such as 35, 37-40.
Thank you for the insightful comment. Following the reviewer's suggestion, I have revised the following references in accordance with the journal guidelines. We have added the number of pages, report numbers, and access dates for references with some information missing.
- Japan, C.O.o. Guide to Business Continuity for Local Public Entities During Earthquake Occurrence and Its Commentary, 1st Edition; 2010; p. 24~36. (line 556)
- Wong, W.N.Z.Z.; Shi, J. Business continuity management system: A complete guide to implementing ISO 22301; Kogan Page Publishers: 2014; pp. 1-273. (line 564)
- Yun, H.S. Final Report on the Development of a Disaster Risk Reduction Activity Management System Support Program and Management Technology; MPSS-Nature-2015-80; Ministry of the Interior and Safety: Seoul, 2018; p. 931. (line 568)
- CISA. National Critical Functions Set. Available online: https://www.cisa.gov/national-critical-functions-set (accessed on 2023.10.21). (line 570)
- Juseong, H.; Wontae, L.; Sunhee, C.; Dongi, S.; Woogi, L. Research on Architecture Policy for Knowledge Informatization - A Study on the Development Strategy of a Government Integrated Computer Center Based on EA (Enterprise Architecture); 9788982424205; Gwacheon: Information and Communication Policy Institute: 2008; p. 188. (line 572)
- Gøtze, J. European Interoperability Reference Architecture. Available online: https://eavoices.com/2015/07/24/european-interoperability-reference-architecture/ (accessed on 2023.10.21). (line 575)
2、The format of formula 2 (line 303) and formula 3 (line 341) in the text should be improved.
Our response: Thank you for your good comment. I have added explanations for each parameter in equations (1) to (3) as follows.
(line 280) Cronbach’s alpha was calculated using Equation (1), where α represents Cronbach’s alpha, N is the total number of items, is the variance for each item, is the variance of the overall score, denotes the variance of a given item i, indicating how much the scores for that item differ, and is the variance of total scores across all items, reflecting the overall score variation.
(line 312) Hence, we adopted Cramer’s V to analyse simple and general correlations between categorical variables without considering ranking information. Cramer’s V was calculated using Equation (2), where is a coefficient that measures the strength of the association between two categorical variables. is the chi-squared statistic for association, is the total sample size, and is the number of categories.
(line 362) The criticalities of the universally validated national functions were calculated using Equation (3). Here, the severity score C was calculated using Nn, the count of respondents who identified the nth national function as critical out of 1,506 participants. The score gauged the urgency of each national function by applying weights to the responses and dividing it by the total number of 1,506 respondents, thus encapsulating the overall severity for the prioritisation of national functions.
3、From section 3.2, it is known that the main source of data for the authors' study is 2018. The global epidemic was used as the background of the study, so should the data after the global epidemic be studied again.
Our response: Thank you for your good comment. I have added explanations for future studies in 5. Conclusions.
(line 468) Third, we plan to compare the pre-pandemic data from this study with post-pandemic data in future research. This comparison will reveal how national function priorities shifted because of the pandemic and help identify new disaster management trends and challenges post-pandemic, thereby enhancing our understanding of effective response strategies in an evolving global context.
4、The population of the survey study came from different regions of the author's country and whether they were public servants or not, so should other conditions such as occupation, income, and education level of the survey population be considered.
Our response: Thank you for your good comment. The current study primarily focuses on the general public and government officials across various regions, emphasizing the universal applicability of the criticality of national functions. Future research could benefit from a more detailed approach that includes socio-economic factors. This would allow for a deeper understanding of national function priorities during disasters. Such an approach would enhance our knowledge of how different social strata perceive the importance of national functions, which could influence disaster management strategies and policies. Regarding this, the additional content added to the main text is as follows.
(line 474) Lastly, future studies should include not only diverse populations from different regions and public officials but also consider additional demographic factors such as occupation, income, and education level to gain more detailed insights into how societal segments perceive the importance of national functions, which may vary according to socioeconomic background.
5、Should V in section 3.4 be in italics.
Our response: Thank you for this comment. I've made the changes you advised in italics. In addition to the points you mentioned, I have also changed the letter V to italic in 24 places.
(line 299) Analysis of the Association Between Regional Differences and National Function Criticality Using Cramer’s V
6、The text size and style in Figures 2 and 3 in the text should be uniform.
Our response: Thank you for your good comment. The font sizes within Figures 1, 2 and 3 have been adjusted to be similar to the main text (size 10) and the font style has been unified to Palatino Linotype. The revised Figures 2 and 3 are as follows.
Figure 1. Research Flow for Deriving the Criteria of Major National Functions’ Recovery Priority (line 161)
|
Figure 2. Regional Distribution of the 1,506 Survey Respondents (line 247) |
Figure 3. Ranking Results of National Function Criticality (line 394)
7、The font size of the formulas in the text is not consistent.
Our response: Thank you for this comment. I edited to match the formulas font size and formatting.
(line 295) (1)
(line 320) (2)
(line 370) (3)
8、The authors mainly rank state function recovery and do not incorporate the rank of disaster occurrence. The ranking of disaster occurrence and the ranking of state function recovery are also different.
Our response: Thank you for your insightful comments. Generally, the ranking of disaster occurrences varies depending on the characteristics and infrastructure levels of each region. Additionally, this paper includes research on establishing a recovery plan and optimizing fund investments for post-disaster recovery. Therefore, our study focused on functional recovery ranking, considering investment optimization and broad applicability. The section you referred to has been appropriately revised and included in the text as a part of our future plan.
(line 478) In addition to demographic factors, a statistical analysis of the occurrence rankings of previous disasters should also be included.
9、The research result is a research study done before the global epidemic, and whether its findings are verified during the epidemic.
Thank you for your good comment. I have added explanations for future studies in 5. Conclusion.
(line 468) Third, we plan to compare the pre-pandemic data from this study with post-pandemic data in future research. This comparison will reveal how national function priorities shifted because of the pandemic and help identify new disaster management trends and challenges post-pandemic, thereby enhancing our understanding of effective response strategies in an evolving global context.
10、The authors' study of disasters focuses on epidemics, whether they consider the order of restoration of state functions in larger natural disasters such as earthquakes, tsunamis or states of war.
Thank you for your good comment. I have added explanations for future studies in 5. Conclusion. We have added a plan to consider larger-scale disasters.
(line 468) Third, we plan to compare the pre-pandemic data from this study with post-pandemic data in future research. This comparison will reveal how national function priorities shifted because of the pandemic and help identify new disaster management trends and challenges post-pandemic, thereby enhancing our understanding of effective response strategies in an evolving global context. Moreover, in addition to global pandemics, it is necessary to consider scenarios such as earthquakes, tsunamis, and states of war.

Round 2
Reviewer 1 Report
Comments and Suggestions for Authors
The manuscript provides detailed answers to the questions raised and is recommended for employment.
Reviewer 2 Report
Comments and Suggestions for Authors
My comments are considered in the revised manuscript. Therefore, it can be accepted in the current form.
Reviewer 3 Report
Comments and Suggestions for Authors
Modified, the quality of the article has been improved very well, it is recommended to receive.
Comments on the Quality of English Languagegood
Reviewer 4 Report
Comments and Suggestions for Authors
This manuscript can be accepted in present form.
Comments on the Quality of English LanguageThis manuscript can be accepted in present form.